# Dual hydrogen production from electrocatalytic water reduction coupled with formaldehyde oxidation via a copper-silver electrocatalyst

Guodong Li[1], Guanqun Han[1], Lu Wang[2], Xiaoyu Cui[1], Nicole K. Moehring[3,4,5], Piran R. Kidambi[3,4,5], De-en Jiang[2,3] ✉ & Yujie Sun[1] ✉

The broad employment of water electrolysis for hydrogen ($H_2$) production is restricted by its large voltage requirement and low energy conversion efficiency because of the sluggish oxygen evolution reaction (OER). Herein, we report a strategy to replace OER with a thermodynamically more favorable reaction, the partial oxidation of formaldehyde to formate under alkaline conditions, using a $Cu_3Ag_7$ electrocatalyst. Such a strategy not only produces more valuable anodic product than $O_2$ but also releases $H_2$ at the anode with a small voltage input. Density functional theory studies indicate the $H_2C(OH)O$ intermediate from formaldehyde hydration can be better stabilized on $Cu_3Ag_7$ than on Cu or Ag, leading to a lower C-H cleavage barrier. A two-electrode electrolyzer employing an electrocatalyst of $Cu_3Ag_7(+)||Ni_3N/Ni(-)$ can produce $H_2$ at both anode and cathode simultaneously with an apparent 200% Faradaic efficiency, reaching a current density of 500 mA/cm$^2$ with a cell voltage of only 0.60 V.

Hydrogen ($H_2$) is not only an important feedstock in the chemical industry (e.g., petroleum refining, ammonia production from the Haber–Bosch process) but also plays an important role in the future energy economy because $H_2$ is a carbon-zero energy carrier and can be directly utilized as a fuel in hydrogen fuel cells[1–3]. Even though mature industry processes exist for the production of $H_2$, such as steam methane reforming, their strong dependence on unsustainable fossil resources and large $CO_2$ emission call on greener alternative methods for producing $H_2$[4]. Against this backdrop, electrocatalytic water splitting, which consists of the $H_2$ and $O_2$ evolution reactions (HER and OER), has attracted worldwide attention these years[5]. Despite the great efforts devoted to developing competent electrocatalysts for both half-reactions and ingenious designs of electrolyzers, the thermodynamics of water splitting dictates its large cell voltage input >1.23 V

(Fig. 1a) and hence high-energy consumption (>4.5−6 kWh/m$^3$ $H_2$)[6–9]. In fact, the more energy-demanding OER produces $O_2$ as a low-value product at the anode while $H_2$ is only produced at the cathode. Therefore, there is an increasing interest in exploring alternative oxidation reactions to replace OER with lower energy input[10]. Even more desirable is that value-added products instead of $O_2$ could be simultaneously obtained at the anode.

Indeed, electrocatalytic oxidation of a variety of inorganic and organic feedstocks has been explored to integrate with HER to produce $H_2$ with lower voltage input[11,12]. For instance, our group has reported several electrocatalytic systems to couple the oxidative valorization of various biomass-derived intermediates (e.g., 5-hydroxymethylfurfural (HMF), furfural, etc.) with HER in aqueous media[13–17]. Other small molecules, such as urea[18,19], ammonia[20], and

[1]Department of Chemistry, University of Cincinnati, Cincinnati, OH 45221, USA. [2]Department of Chemistry, University of California Riverside, Riverside, CA 92521, USA. [3]Department of Chemical and Biomolecular Engineering, Vanderbilt University, Nashville, TN 37212, USA. [4]Interdisciplinary Graduate Program in Materials Science, Vanderbilt University, Nashville, TN 37235, USA. [5]Vanderbilt Institute of Nanoscale Science and Engineering, Nashville, TN 37212, USA. ✉e-mail: de-en.jiang@vanderbilt.edu; yujie.sun@uc.edu

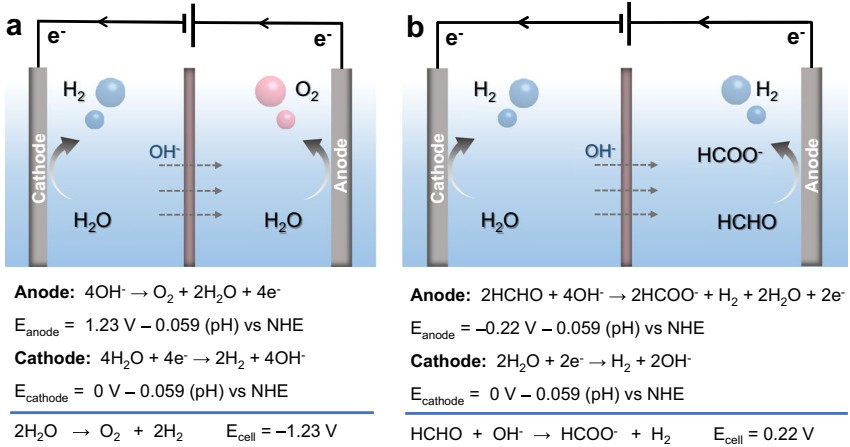

**Fig. 1 | Schematic illustration of water electrolysis and HCHO electrooxidation. a** Conventional electrocatalytic water splitting under alkaline conditions.
**b** Electrocatalytic water reduction coupled with HCHO oxidation under alkaline conditions.

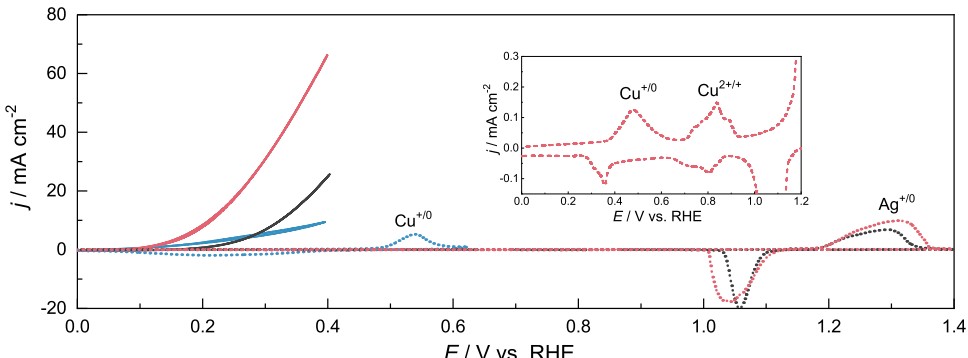

**Fig. 2 | HCHO electrooxidation over Cu, Ag, and CuAg catalysts on RDE.** CV curves of Cu/RDE (blue), Ag/RDE (black), and $Cu_3Ag_7$/RDE (red) in 1.0 M KOH in the absence (dashed) and presence (solid) of 0.6 M HCHO collected at 1500 rpm and 10 mV/s. Inset shows the expanded CV of copper oxidation on $Cu_3Ag_7$/RDE.

hydrazine[21], have also been reported to replace OER for more energy-efficient $H_2$ production from water, which produces $CO_2$ and/or $N_2$ at the anode. Despite the great progress in this direction, $H_2$ is only produced at the cathode and most electrocatalytic systems still require a cell voltage larger than 1 V to reach an industrially relevant current density (>500 mA/cm²). Furthermore, the utilization of biomass-derived feedstocks such as HMF and furfural for the large-scale production of $H_2$ is still questionable because of the tremendous disparity in their scalability and the future $H_2$ demand. Consequently, it remains a great challenge to develop an alternative strategy to produce $H_2$ from water with ultra-low voltage input, and even more desirable is to produce $H_2$ at both cathode and anode.

We were inspired by the advance in $H_2$ generation from liquid-phase hydrogen storage molecules, such as $NaBH_4$[22], $NH_3BH_3$[23,24], and HCOOH[25,26], which are able to release $H_2$ under thermocatalytic conditions. Among those liquid organic hydrogen storage molecules, formaldehyde (HCHO) is particularly appealing, because the partial HCHO oxidation (FOR) to formate is also able to release $H_2$ at the anode with a very small thermodynamic potential ($HCHO + 2OH^- \rightarrow HCOO^- + 1/2H_2 + H_2O + e^-$, $E = -0.22$ V vs. RHE, Fig. 1b)[27]. Furthermore, HCHO is a low-cost chemical feedstock with a gigantic annual yield while its oxidation product formate (and later formic acid) is a more valuable chemical[28,29]. Finally, coupling FOR with HER for $H_2$ production may also provide environmental benefits if toxic formaldehyde residues in wastewater pollutants could be adopted as the feedstock.

Even though $H_2$ generation from the electrochemical oxidation of HCHO has been reported on a few monometallic electrodes[30–32], it remains less explored to couple it with water reduction for dual $H_2$

production at both anode and cathode. Herein, we report a novel inexpensive electrocatalytic system, $Cu_3Ag_7$ and $Ni_3N$/Ni as the anode and cathode electrocatalysts, respectively, to drive FOR and HER under alkaline conditions, which can produce $H_2$ with an apparent 200% Faradaic efficiency and reach industrially relevant current densities of 100 and 500 mA/cm² at cell voltages of only 0.22 and 0.60 V, respectively. Overall, the energy consumption of our two-electrode electrolyzer for $H_2$ production is merely 0.30 and 0.70 kWh/m³ $H_2$ at current densities of 100 and 500 mA cm⁻², respectively, much lower than the theoretical energy demand of overall water splitting electrolysis (2.93 kWh/1 m³ $H_2$).

## Results and discussion

### Screening of catalysts for formaldehyde oxidation

Inexpensive nitrate salts of copper and silver were utilized as the feedstocks to first prepare the corresponding monometallic electrocatalysts on glassy carbon rotating disk electrode (RDE) via a facile electrodeposition method (see the "Methods" section for details). As shown in Fig. 2, in the absence of HCHO, Cu/RDE showed a well-defined oxidation feature from $Cu^0$ to $Cu^+$ at ca. 0.5 V vs. RHE in 1.0 M KOH; while the oxidation peak of $Ag^0$ to $Ag^+$ from Ag/RDE did not appear until -1.2 V vs. RHE. Upon the addition of 0.6 M HCHO, an apparent anodic current rise was observed beyond the onset potential (defined at 0.1 mA cm⁻²) of 0.05 V vs. RHE on Cu/RDE and a current density of 10 mA/cm² was obtained at 0.4 V vs. RHE, indicating the electrochemical oxidation of HCHO on Cu/RDE. Despite the more positive onset potential (0.2 V vs. RHE) for HCHO oxidation, a much more rapid current rise was observed on Ag/RDE, which was able to produce an

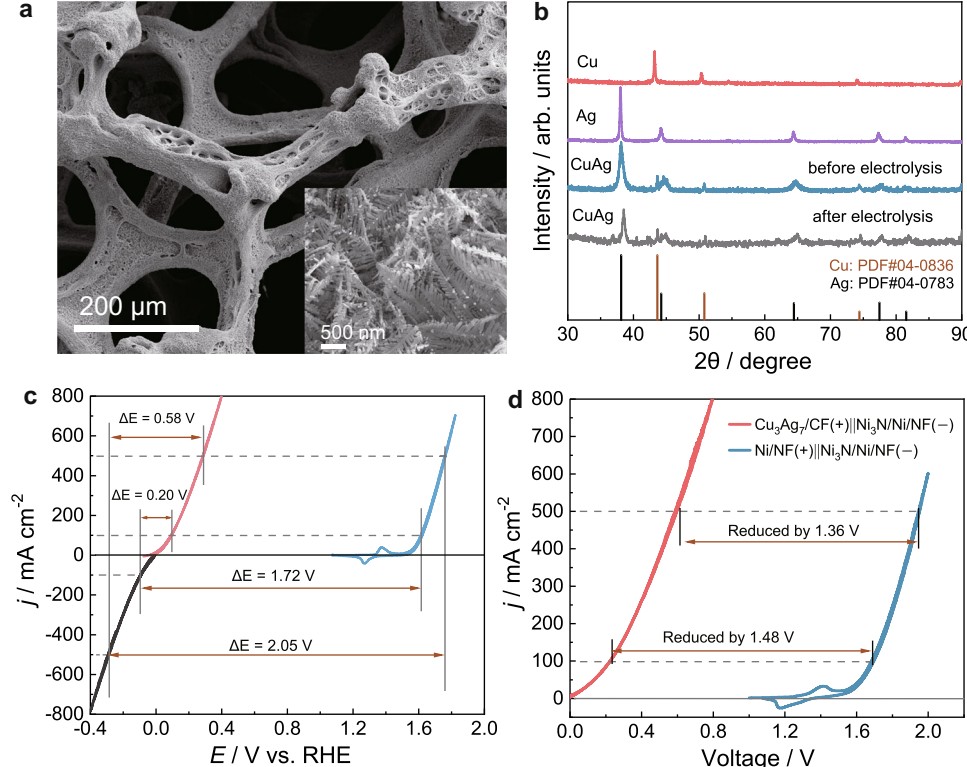

**Fig. 3 | Characterizations and HCHO oxidation performances of Cu₃Ag₇/CF electrocatalysts. a** SEM image of as-prepared Cu₃Ag₇/CF. **b** XRD patterns of Cu/CP, Ag/CP, and Cu₃Ag₇/CP prior to and post electrolysis. **c** CV curves of Cu₃Ag₇/CF for HCHO oxidation (red) in 1.0 M KOH with 0.6 M HCHO, Ni₃N/Ni/NF for HER (black), and Ni/NF for OER (blue) in 1.0 M KOH collected at 10 mV/s. Voltage gaps to reach 100 and 500 mA/cm² are indicated. **d** The two-electrode CV curves of HER/FOR (red) and HER/OER (blue) collected at 10 mV/s, in which Cu₃Ag₇/CF and Ni₃N/Ni/NF were employed as the anode and cathode for the former while Ni/NF and Ni₃N/Ni/NF for the latter. For FOR, the anolyte was 1.0 M KOH and 0.6 M HCHO while for all the other conditions, 1.0 M KOH was the electrolyte.

anodic current density of 26 mA/cm² at 0.4 V RHE. The differences in onset potential and the anodic current rise of HCHO oxidation on Cu/RDE versus Ag/RDE prompted us to finely tune the ratio between Cu and Ag precursors in electrodeposition, aiming to synthesize a bimetallic CuAg electrocatalyst with optimized activity.

As shown in Supplementary Fig. 1, by systematically varying the Cu/Ag ratio from 9/1 to 1/9 (see the Supplementary Information for electrodeposition details), all the bimetallic CuAg/RDE electrodes possess similar electrochemical double layer capacitances ($C_{dl}$ = 0.7−0.8 mF/cm²), close to those of the pristine monometallic samples (0.5−0.6 mF/cm²). Therefore, these bimetallic CuAg/RDE samples have similar electrochemical active surface area, which allows a fair comparison of their electrocatalytic activity of HCHO oxidation based on the measured geometric current density. Supplementary Fig. 2a plots the cyclic voltammograms (CVs) of all the CuAg/RDE samples as well as Cu/RDE and Ag/RDE measured under the same condition (0.6 M HCHO in 1.0 M KOH). It is apparent that Cu₃Ag₇/RDE exhibited the best performance (Supplementary Fig. 2b). The ICP-MS analysis result of Cu₃Ag₇/RDE confirmed the Cu/Ag ratio was 3:7. Therefore, all the subsequent electrochemical HCHO oxidation studies were carried out using Cu₃Ag₇ as the optimal electrocatalyst.

In the absence of HCHO, Cu₃Ag₇/RDE still retained the typical copper redox features of Cu⁺/⁰ and Cu²⁺/⁺ between 0.4 and 1.0 V vs. RHE but with suppressed current density (Fig. 2 inset) relative to that of Cu/RDE in 1.0 M KOH. Further positive scanning revealed an analogous oxidation feature of Ag⁰ to Ag⁺ beyond 1.2 V vs. RHE as observed for Ag/RDE. The addition of 0.6 M HCHO resulted in a drastic anodic current rise starting at ca 0.1 V vs. RHE, which was able to reach 66 mA/cm² at 0.4 V vs. RHE (Fig. 2), substantially higher than those obtained on monometallic Cu/RDE (10 mA/cm²) and Ag/RDE (26 mA/cm²) at the same potential.

Using Cu₃Ag₇/RDE as a model electrode, we investigated the impact of HCHO concentration on the obtained current density (Supplementary Fig. 3) and it was found that 0.6 M HCHO produced the maximum current density at 0.4 V vs. RHE in 1.0 M KOH. Increasing the HCHO concentration beyond 0.6 M resulted in decreased anodic current, likely due to more disproportionation of HCHO at higher concentrations (Cannizzaro reaction)[33]. Next, the impact of OH⁻ concentration was also probed and the results were compiled in Supplementary Fig. 4. In 1.0 M NaClO₄, no anodic current was observed on Cu₃Ag₇/RDE between 0 and 0.4 V vs. RHE when 0.1 M HCHO was added. However, a slight increase to 1 mM [OH⁻] was able to produce a noticeable anodic current at 0.4 V RHE. A higher HCHO oxidation current was observed along the increase in [OH⁻]. Overall, these results demonstrate that an alkaline condition is necessary for the electrochemical oxidation of HCHO on Cu₃Ag₇/RDE.

## Electrolysis performances

With the optimal Cu/Ag ratio in hand, we next sought to electrodeposit Cu₃Ag₇ on highly porous and conductive substrates for electrolysis studies. To avoid the introduction of other metal composites, commercially available copper foam was adopted as the catalyst support and current collector. A slightly modified electrodeposition approach was utilized to prepare Cu₃Ag₇ on copper foam (CF) and the resulting electrode was named Cu₃Ag₇/CF (see the "Methods" section for details). The scanning electron microscopy (SEM) image of a pristine copper foam indicates its porous skeleton with a smooth surface (Supplementary Fig. 5a). After the electrodeposition of Cu₃Ag₇, the as-prepared Cu₃Ag₇/CF shows nearly complete coverage of the copper foam by pine needle-like electrodeposits (Fig. 3a and Supplementary Fig. 5b). Energy-dispersive X-ray spectroscopy analysis (EDX) of Cu₃Ag₇/CF resulted in a Cu/Ag ratio very close to 3:7 (Supplementary

Fig. 5c, d), in agreement with the ICP-MS results of $Cu_3Ag_7$/RDE. To shed light on the crystallinity of electrodeposited $Cu_3Ag_7$, X-ray diffraction was performed. In this case, carbon paper (CP), instead of copper foam, was used as the catalyst support, in order to avoid the interference of the background signal from the copper foam. The obtained XRD patterns of Cu/CP, Ag/CP, and $Cu_3Ag_7$/CP included in Fig. 3b confirmed that the bimetallic sample consists of both crystalline Cu (JCPDS Card no. 04-0836) and Ag (JCPDS Card no. 04-0783)[34,35]. The most prominent peaks at around 44° and 38° are corresponding to the (111) facets of Cu and Ag, respectively, which are preserved in $Cu_3Ag_7$/CP. To further investigate the structure of $Cu_3Ag_7$/CF, high-resolution transmission electron microscopy (HRTEM) combined with selected area electron diffraction (SAED) was performed. Supplementary Fig. 6a presents a clear dendritic structure in line with the SEM images (Fig. 3a and Supplementary Fig. 5c). The HRTEM images (Supplementary Fig. 6b, c) show the well-resolved lattice fringes with an inter-planar distance of 0.237 and 0.208 nm corresponding to the (111) crystal planes of cubic Ag and Cu, respectively. The distinct diffraction rings from SAED (Supplementary Fig. 6d) demonstrate the polycrystalline nature and could be indexed to the (111), (200), (220), and (311) planes of Ag and Cu[36], respectively, in good agreement with the XRD results. High-angle-annular dark-field STEM (HAADF-TEM) and EDX element mapping images reveal a uniform distribution of Cu and Ag throughout the dendrites (Supplementary Fig. 6e).

Next, a two-compartment electrochemical cell (H-cell) with an anion exchange membrane was used to investigate the electrochemical HCHO oxidation on $Cu_3Ag_7$/CF. As shown in Fig. 3c, the CV curve of $Cu_3Ag_7$/CF collected in 1.0 M KOH with 0.6 M HCHO presents a nearly zero onset potential and very steep anodic current increase, reaching 100 and 500 mA/cm² at merely 0.10 and 0.28 V vs. RHE, respectively. In sharp contrast, when electrodeposited nickel nanoparticles on nickel foam (Ni/NF) were employed as the OER electrocatalysts, much higher potentials (1.62 and 1.76 V vs. RHE) were required to reach the same current densities (100 and 500 mA/cm²). We previously reported an excellent earth-abundant HER electrocatalyst composed of interfaced $Ni_3N$ and Ni on nickel foam ($Ni_3N$/Ni/NF) in 1.0 M KOH[37]. For the sake of comparison, we also included the CV curve of $Ni_3N$/Ni/NF for the cathodic $H_2$ evolution in Fig. 3c. With the above CVs, we were able to deduce that the voltage inputs for HER coupled with FOR would be much smaller to reach industrially relevant current densities like 100 and 500 mA/cm², requiring only 0.20 and 0.58 V, respectively, when $Cu_3Ag_7$/CF and $Ni_3N$/Ni/NF were employed as the anode and cathode electrocatalysts, respectively. However, much larger voltages were demanded (>1.7 V) for water splitting (HER/OER) to produce the same current density when using the Ni/NF (anode) and $Ni_3N$/Ni/NF (cathode) catalyst couple. Indeed, as plotted in Fig. 3d, when a two-electrode electrolyzer of FOR/HER was assembled using $Cu_3Ag_7$/CF as the anode and $Ni_3N$/Ni/NF as the cathode, the corresponding CV curve collected in 1.0 M KOH with 0.6 M HCHO only required cell voltages of 0.22 and 0.60 V to deliver the current densities of 100 and 500 mA cm⁻², respectively, yet much higher cell voltages of 1.70 and 1.96 V were needed for conventional water splitting electrolysis. In short, this FOR/HER strategy using the $Cu_3Ag_7$/CF(+)‖$Ni_3N$/Ni/NF(−) electrode couple exhibited great superiority compared to reported systems of HER integrated with the oxidation of various inorganic and organic feedstocks (Table S1), in terms of applied voltage and Faradaic efficiency for $H_2$ production.

## Products analysis

Since the commercially purchased HCHO (37 wt% in $H_2O$) contains methanol as a stabilizer and HCHO oxidation will produce formate under alkaline conditions, it is of critical importance to determine if $Cu_3Ag_7$/CF can catalyze the oxidation of methanol and formate within this small potential window. CV of $Cu_3Ag_7$/RDE from 0 to 0.4 V vs. RHE in 1.0 M KOH was first collected after the addition of 0.1 M formic acid,

0.1 M methanol, and 0.1 M HCHO, respectively. As shown in Supplementary Fig. 7, the increase in anodic current on $Cu_3Ag_7$/RDE was negligible upon the addition of either methanol or formic acid. However, a rapid anodic current rise was observed once HCHO was added, indicating that $Cu_3Ag_7$/RDE could effectively catalyze the electrooxidation of HCHO but not formic acid or methanol within this potential region, likely due to the smaller bond dissociation energy of C−H bond in H·CHO (88.0 ± 0.2 kcal mol⁻¹) compared to all the bonds in methanol and formic acid[38].

Figure 4a presents the chronoamperometry curve of $Cu_3Ag_7$/CF in 1.0 M KOH upon the continuous addition of 0.1 M formic acid, 0.1 M methanol, and 0.1 M HCHO. It is apparent that negligible anodic current increase was observed upon the addition of either formic acid or methanol with a voltage input of 0.6 V, indicating that $Cu_3Ag_7$/CF was not able to catalyze their electrochemical oxidation with this small voltage. In contrast, the addition of 0.1 M HCHO led to an immediate anodic current rise to 230 mA/cm². These results prove that $Cu_3Ag_7$/CF possesses an excellent selectivity towards the electrochemical oxidation of HCHO with low voltage input, which is not affected by the presence of methanol and formate.

Gas chromatography was also performed to confirm (Supplementary Fig. 8) and quantify the amount of produced gas from the H-cell where $Cu_3Ag_7$/CF was employed as the anode and $Ni_3N$/Ni/NF as the cathode in a two-electrode configuration. As shown in Fig. 4b, without applied voltage, no $H_2$ was detected, suggesting that $Cu_3Ag_7$/CF was not able to thermocatalytically drive HCHO oxidation in 1.0 M KOH with 0.6 M HCHO at room temperature. Along with the increase of voltage input from 0.2 to 0.8 V, because of the faster $H_2$ production rate, continuously decreased electrolysis time was needed to produce 2 mmol $H_2$ from the anode chamber. Comparing the experimentally measured amount of $H_2$ and the calculated amount of $H_2$ based on passed charge during each electrolysis of different voltage inputs (0.2–0.8 V) confirmed that 100% Faradaic efficiency (FE) was achieved for anodic $H_2$ production within the entire voltage range (Fig. 4b inset and Supplementary Fig. 9). In the meantime, cathodic $H_2$ production from the $Ni_3N$/Ni/NF cathode was also able to produce $H_2$ with a 100% FE. For instance, Fig. 4c overlaps the measured $H_2$ amounts over the theoretical $H_2$ amounts along the passed charge for both cathode and anode chambers during electrolysis carried out at 0.8 V when 1.0 M KOH with 0.6 M HCHO was used as the anolyte and 1.0 M KOH as the catholyte. The nearly perfect alignment of the $H_2$ amounts for both chambers confirmed that 100% FE of $H_2$ production was realized for both electrodes and collectively 200% FE for overall $H_2$ generation.

The organic products in the liquid phase of the anode chamber were determined and quantified to assess the carbon balance of electrolysis. The amounts of methanol and formate were determined from ¹H NMR measurements using *t*-butanol as an internal standard (Supplementary Fig. 10). Besides the quantity resulting from the Cannizzaro reaction, all the additional formate was produced from the electrochemical HCHO oxidation (Supplementary Fig. 11 and Table S2). As shown in Fig. 4d, 100% FE of formate production was achieved for five consecutive 1 h controlled-current electrolysis (150 mA) using the $Cu_3Ag_7$/CF and $Ni_3N$/Ni/NF couple but fresh electrolyte for each cycle. In the meantime, 200% FE of overall $H_2$ production was retained for all five cycles as well. Further, the remaining HCHO was quantified via UV−vis absorption measurement following the Hantzsch reaction (see the "Methods" section for details)[39], using a pre-established calibration curve (Supplementary Fig. 12). From the results (Supplementary Fig. 13 and Table S2) we conclude that 100% carbon balance of HCHO was maintained for the above five consecutive electrolysis cycles using the same electrode couple.

The stability of the $Cu_3Ag_7$/CF and $Ni_3N$/Ni/NF electrode couple was further tested by both chronopotentiometry and chronoamperometry electrolysis. Figure 4e shows nearly identical chronopotentiometric curves for five consecutive controlled-current

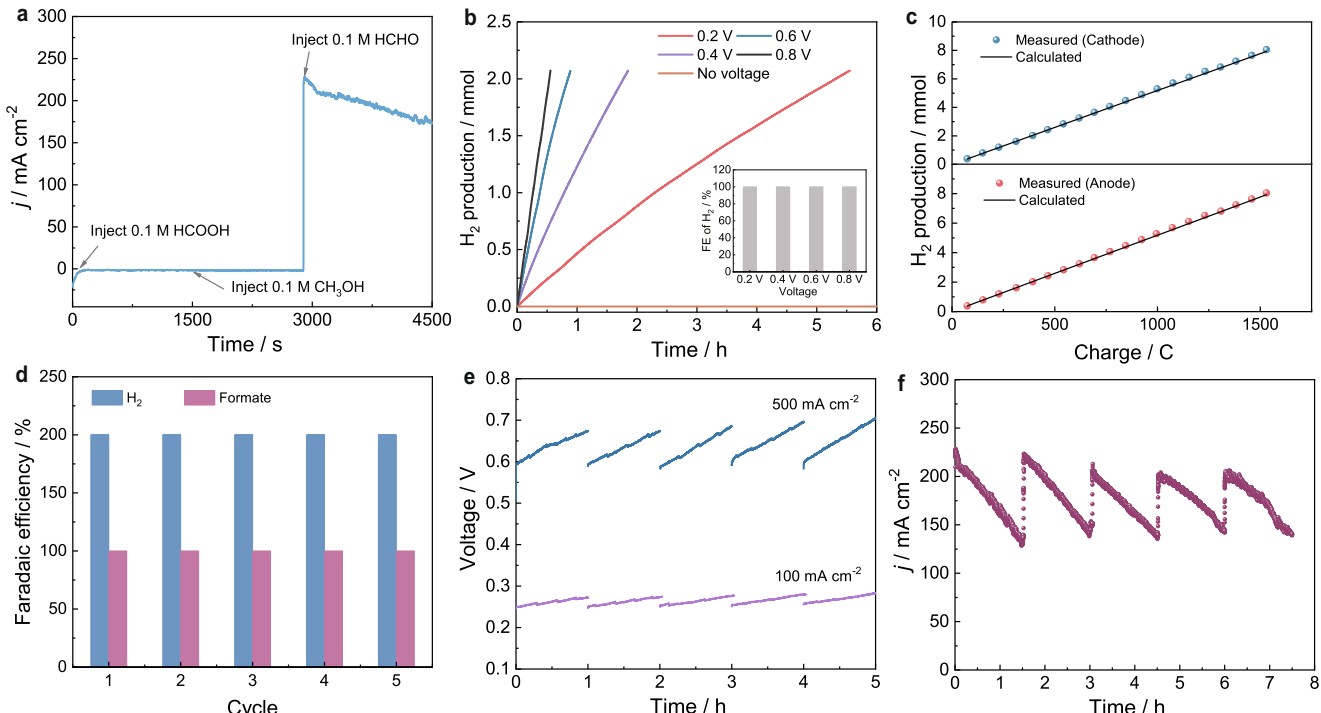

**Fig. 4 | Dual hydrogen production of FOR/HER system.** The electrolysis experiments were conducted in a two-electrode electrolyzer using Cu₃Ag₇/CF anode and Ni₃N/Ni/NF cathode. **a** Chronoamperometric curve collected at a cell voltage of 0.6 V in 1.0 M KOH with the continuous addition of 0.1 M HCOOH, 0.1 M CH₃OH, and 0.1 M HCHO in the anode chamber. **b** Comparison of the experimentally measured amount of H₂ from the anode chamber with different voltage inputs. Inset shows the Faradaic efficiency of H₂ production. **c** Comparison of the experimentally measured H₂ amounts with the theoretical H₂ amounts calculated from the passed charge for both cathode and anode chambers during electrolysis at a cell voltage of 0.6 V. **d** Faradaic efficiencies of H₂ and formate production for five consecutive 1 h controlled-current (150 mA) electrolysis cycles. **e** Chronopotentiometric curves for five consecutive controlled-current electrolysis cycles conducted at 100 and 500 mA/cm². **f** Chronoamperometric curves at a cell voltage of 0.6 V with the periodic replenishment of fresh HCHO back to its original 0.1 M concentration in anolyte.

electrolysis conducted at 100 and 500 mA/cm² in a fresh anolyte (1.0 M KOH and 0.6 M HCHO) of each cycle using the same electrode couple. In addition, a 7.5 h chronoamperometry experiment was carried out in the same electrolyte at a cell voltage of 0.6 V with the periodic replenishment of fresh HCHO back to its original 0.1 M concentration. Immediate resume of the anodic current was observed for each period (Fig. 4f). Post-electrolysis characterization of Cu₃Ag₇/CF displayed negligible changes in morphology, crystallinity, and electrochemical double-layer capacitance (Fig. 3b and Supplementary Figs. 14, 15). Overall, these results corroborate the superior structural robustness and mechanical stability of Cu₃Ag₇/CF for long-term electrochemical HCHO oxidation for anodic H₂ production with low voltage input.

## Theoretical computations
To shed light on the improved performance of the bimetallic Cu₃Ag₇/CF relative to monometallic Cu/CF and Ag/CF in the anodic H₂ production from electrocatalytic HCHO oxidation, density functional theory (DFT) computation was performed to examine the structure and energetics of key steps during FOR. Because only a small voltage is needed to drive FOR, one can assume that the electrocatalytic reaction proceeds close to the thermodynamic limit. Under thermal conditions, the commonly accepted mechanism of FOR is that HCHO first becomes hydrated and deprotonated in the alkaline solution to yield the H₂C(OH)O anion, which then adsorbs on the catalyst surface[40,41]. The resulting H₂C(OH)O* intermediate is dehydrogenated via C–H cleavage to yield HCOOH* and H* (Fig. 5a), which is a key step we focus on. Our experimental characterization showed that the (111) facets are preferentially exposed for the Cu, Ag, and Cu₃Ag₇ catalysts, so we used the (111) surfaces to compare the three catalysts (see Supplementary Fig. 16a for the details of the models). First, we compared the relative adsorption energy of the H₂C(OH)O* intermediate: as shown in Fig. 5b, adsorption on Cu₃Ag₇ is most preferred, with the O group anchored at the Cu₂Ag₁ hollow site; the top view shows a clear staggered conformation of the H₂COH group relative to the Cu₂Ag₁ site (Fig. 5c), distinct from the more eclipsed conformations on Cu(111) and Ag(111) (Supplementary Fig. 16b). Analysis of the projected density of states of the anchored O group of H₂C(OH)O* at the Cu₂Ag₁ site indicates that there are strong orbital mixings of O 2p states with Cu 3d states from −1.5 to −1.0 eV as well as with Ag 4d states from −6 to −5 and −4 to −3 eV (Supplementary Fig. 16c). In other words, the existence of the separate two d-bands on the bimetallic surface provides more flexibility in adsorbing and stabilizing the H₂C(OH)O* intermediate. Indeed, C–H cleavage is much more facile on Cu₃Ag₇ from the staggered adsorption conformation (Fig. 5d). After C–H cleavage, the formation of H₂ from two H* was found to be facile as well on Cu₃Ag₇ (Fig. 5e): the energy barrier is 0.66 eV, which is expected to be further lowered when entropy is taken into account. To complete the reaction, HCOOH* will desorb from the surface and then deprotonate to form formate in the solution.

## Electrocatalytic paraformaldehyde oxidation and energy efficiency analysis
The promising activity of Cu₃Ag₇ towards HCHO oxidation to produce H₂, corroborated by our DFT results, prompted us to further expand the substrate scope. Especially, we were motivated to explore the possibility of utilizing paraformaldehyde as the feedstock for the following reasons. In contrast to HCHO aqueous solutions with limited concentration (e.g., 37 wt%) and containing methanol as the stabilizer, paraformaldehyde is solid under ambient conditions, possessing much higher mass density and also lower cost[42]. It is known that HCHO can

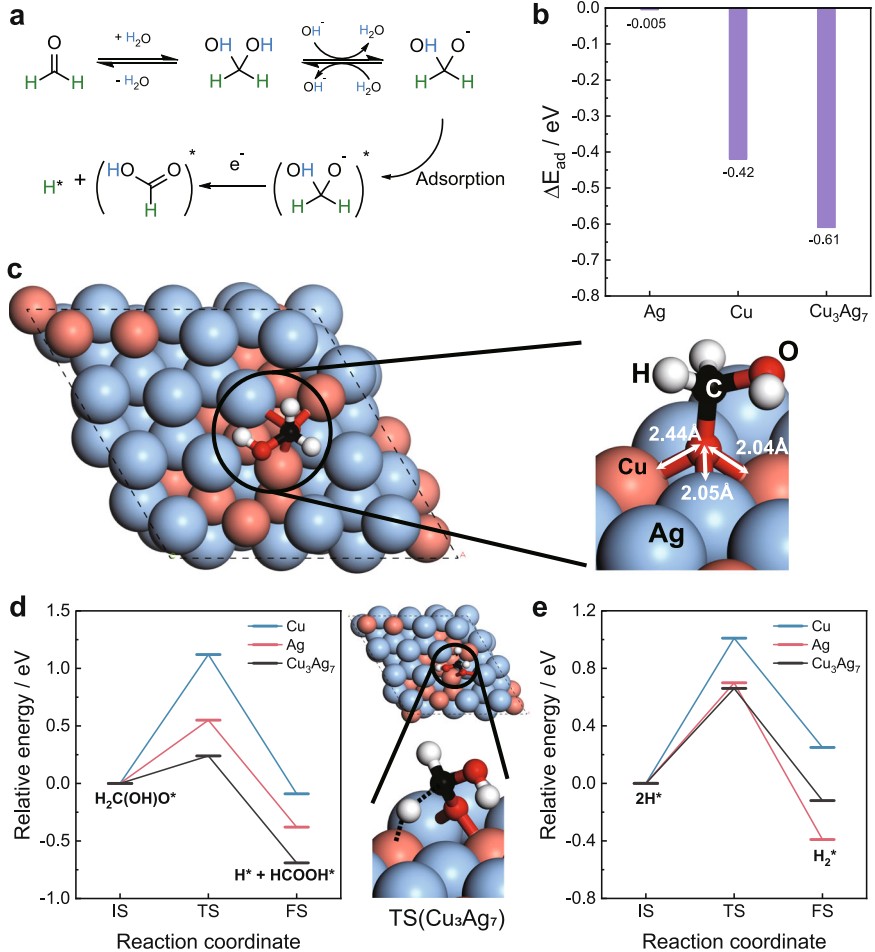

**Fig. 5 | DFT theoretical computations. a** Proposed mechanism of HCHO oxidation to HCOOH. **b** Computed adsorption energy of the $H_2C(OH)O$ intermediate on the three model surfaces. **c** Optimized adsorption geometry of $H_2C(OH)O$ on $Cu_3Ag_7$. **d** Initial (IS), transition (TS), and final (FS) states of $H_2C(OH)O$ dehydrogenation on the three model surfaces, together with the TS structure on $Cu_3Ag_7$. **e** $H_2$ formation via the Tafel step on the three surfaces.

be released from paraformaldehyde when dissolved in an aqueous solution[43]. To our delight, as shown in Supplementary Fig. 17a, $Cu_3Ag_7$/RDE still exhibited excellent electrocatalytic activity for the oxidation of paraformaldehyde, superior to Cu/RDE and Ag/RDE. Concentration dependence experiments indicated that 10.0 g/L paraformaldehyde resulted in the highest oxidation current density (60 mA/cm$^2$) on $Cu_3Ag_7$/RDE in 1.0 M KOH (Supplementary Fig. 17b). Figure 6a compares the CV curves collected on $Cu_3Ag_7$/CF and Ni/NF in the presence and absence of 10.0 g/L paraformaldehyde, respectively. It is apparent that much smaller potentials of 0.13 and 0.36 V vs. RHE are required to produce 100 and 500 mA/cm$^2$, respectively, than those required for OER on Ni/NF. Nearly unity Faradaic efficiencies were obtained for both $H_2$ and formate production when paraformaldehyde was employed as the feedstock and $Cu_3Ag_7$/CF as the anode electrocatalyst at different applied potentials from 0.1 to 0.4 V vs. RHE (Fig. 6b, Supplementary Figs. 8b and 18, and Table S3). Notably, when paraformaldehyde was used as the oxidation substrate, the yields of methanol and formate from the competing Cannizzaro reaction were much lower than those directly using HCHO solutions, suggesting another advantage of using paraformaldehyde as the feedstock for our dual HER strategy. The long-term stability test of electrolysis using paraformaldehyde was also carried out using a flow cell consisting of AEM and Ni$_3$N/Ni/NF(−)||Cu$_3$Ag$_7$/CF(+) electrode couple in a zero-gap configuration. The electrolysis was performed at 100 mA cm$^{-2}$ with 1.0 M KOH as the catholyte and 1.0 M KOH plus 10.0 g/L paraformaldehyde as the anolyte, both of which were fed into the cell at a

flow rate of 50 mL min$^{-1}$. The electrolyte was refreshed every 24 h. Supplementary Fig. 19 demonstrates that our dual $H_2$ production strategy (HER/FOR) using the Ni$_3$N/Ni/NF(−)||Cu$_3$Ag$_7$/CF(+) electrode couple can maintain a low voltage input (0.4–0.6 V without iR correction) to deliver 100 mA cm$^{-2}$ over 260 h, indicating the great robustness of our electrocatalysts for dual $H_2$ production.

From an energy efficiency perspective, our dual $H_2$ production system surpasses those conventional water-splitting systems to a substantial extent. For instance, Fig. 6c compares the electricity consumption between our dual $H_2$ production strategy (HER/FOR) using the Ni$_3$N/Ni/NF(−)||Cu$_3$Ag$_7$/CF(+) electrode couple and traditional water electrolysis (HER/OER) using the Ni$_3$N/Ni/NF(−)||Ni/NF(+) electrode couple. To produce $H_2$ at a current density of 100 mA/cm$^2$, our dual $H_2$ production system only requires 0.30 kWh of electricity per m$^3$ $H_2$ whereas conventional water splitting demands 4.10 kWh. Even at an industrially applicable current density of 500 mA/cm$^2$, the electricity consumption of our dual $H_2$ production system is still as small as 0.70 kWh/m$^3$ $H_2$, much lower than that of water electrolysis (4.70 kWh/m$^3$ $H_2$).

In summary, we report a bimetallic electrocatalyst $Cu_3Ag_7$ for efficient $H_2$ production at the anode of an electrolyzer which integrates the partial oxidation of formaldehyde with water reduction to realize dual HER with an apparent 200% Faradaic efficiency. DFT computation reveals a key adsorption conformation of the $H_2C(OH)O^*$ intermediate on $Cu_3Ag_7$ that is highly conducive to C–H cleavage. In addition to formaldehyde solution, solid-phase paraformaldehyde can be equally

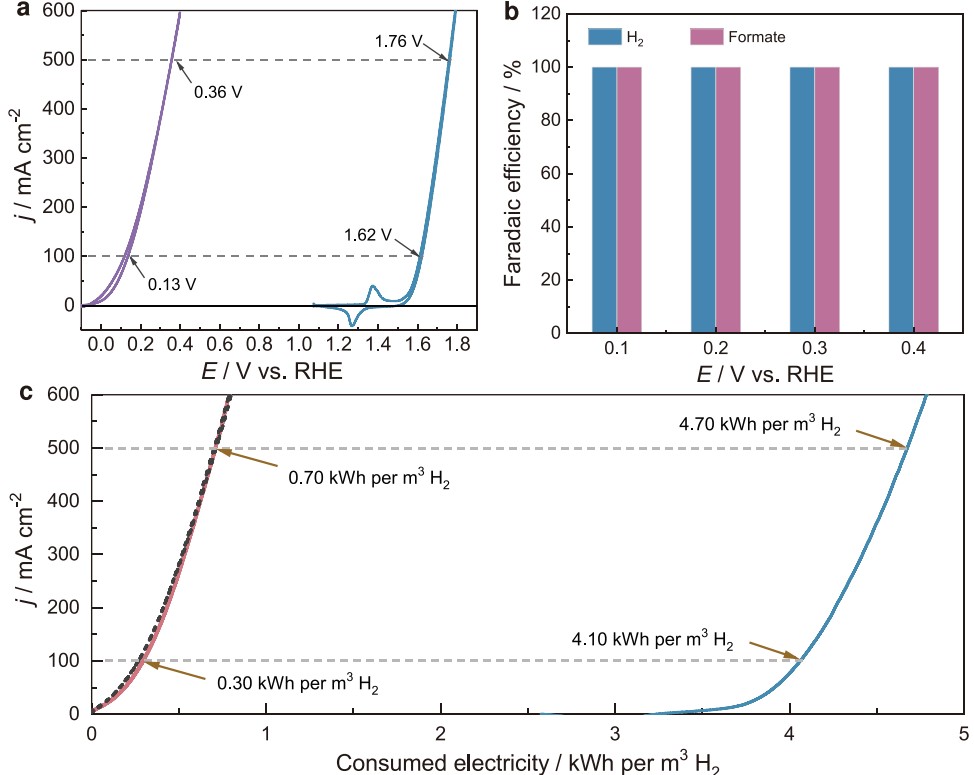

**Fig. 6 | Electrocatalytic paraformaldehyde oxidation and energy efficiency analysis. a** CV curves of Cu₃Ag₇/CF (red) in 1.0 M KOH with 10.0 g/L paraformaldehyde and Ni/NF (blue) in 1.0 M KOH collected at 10 mV/s. **b** Faradaic efficiencies of H₂ and formate production in the anode chamber during each electrolysis at different potentials using Cu₃Ag₇/CF as the anode, 1.0 M KOH as the catholyte, and 1.0 M KOH with 10.0 g/L paraformaldehyde as the anolyte. **c** Comparative analysis of the calculated electricity consumption for H₂ production between our formaldehyde (red) or paraformaldehyde (black) oxidation-integrated strategy using the Ni₃N/Ni/NF(−)||Cu₃Ag₇/CF(+) electrode couple and traditional water electrolysis (HER/OER) using the Ni₃N/Ni/NF(−)||Ni/NF(+) electrode couple.

employed as a reactant to realize similar performance, paving the way for practical application on a large scale.

## Methods

### Chemicals

All chemicals were used as received without any further purification. Copper(II) nitrate hydrate, silver nitrate, copper(II) sulfate pentahydrate, silver sulfate, potassium hydroxide (>85%), formaldehyde solution (37 wt% in H₂O), paraformaldehyde, formic acid (>99%), methanol (>99.9%), tert-butanol (anhydrous), hydrochloric acid (ACS reagent, 37%), sodium citrate were purchased from Sigma Aldrich. Ammonium chloride and nickel(II) chloride hexahydrate were purchased from Alfa Aesar. Copper foam and nickel foam with a purity >99.99% were purchased from MTI. Deionized water (18 MΩ cm) from a Barnstead E-Pure system was used in all experiments.

### Synthesis

**Synthesis of Cu, Ag, and CuAg catalysts on rotating disk electrode.** The catalysts were synthesized through a convenient and straightforward electrodeposition method by using a typical three-electrode system on a VMP-3 potentiostat (Biologic Science Instrument). A rotating disk electrode with glassy carbon (RDE, 0.196 cm⁻², Pine Research Instrumentation) was used as the working electrode with a leakless Ag/AgCl (eDAQ) reference electrode and a Pt mesh counter electrode. −0.2 V vs. Ag/AgCl was applied to the RDE working electrode for the reduction of Cu and Ag ions in the electrolyte. The electrodeposition ended when a charge of 60 mC had been passed. Cu modified on RDE (Cu/RDE) and Ag modified on RDE (Ag/RDE) were electrodeposited at −0.2 V vs. Ag/AgCl from an aqueous electrolyte containing specific amounts of Cu(NO₃)₂ (20.0 mM, 30 mL)

and AgNO₃ (20.0 mM, 30 mL), respectively. For comparison, the CuₓAg₁₀₋ₓ/RDE (x = 1, 3, 5, 7, and 9) were prepared under the same conditions from precursor electrolytes (20.0 mM, 30.0 mL) containing AgNO₃ and Cu(NO₃)₂ solutions with different concentration ratios (from 1/9 to 9/1).

**Synthesis of Cu₃Ag₇ catalysts on Cu foam (Cu₃Ag₇/CF)**
To avoid the introduction of other metal composites, copper foam (0.5 cm × 0.5 cm) was used as the catalyst support and current collector. The copper foam was sequentially washed with 1.0 M HCl, ethanol, and deionized water (each for 10 min). Cu₃Ag₇/CF was synthesized through electrodeposition on a Gamry Interface 1000 electrochemical workstation with a three-electrode configuration, using a carbon rod and a leakless Ag/AgCl (eDAQ) electrode as the counter electrode and reference electrode, respectively. Cu₃Ag₇/CF was electrodeposited at a constant current of −3.0 A cm⁻² for 30 s from an aqueous electrolyte (30 mL) containing specific amounts of Ag₂SO₄ (14.0 mM) and CuSO₄ (6.0 mM) in 1.5 M H₂SO₄ under Ar without stirring. In addition, 0.1 M sodium citrate was added to the precursor as a complexing agent. To avoid the interference of the background signal from copper foam in the XRD tests, we prepared Cu-modified, Ag-modified, and Cu₃Ag₇-modified carbon paper electrodes (Cu/CP, Ag/CP, and Cu₃Ag₇/CP) samples following the same preparation conditions. The only difference is that copper foam was replaced by carbon paper.

**Synthesis of Ni/NF and Ni₃N/Ni/NF**
Based on our previous work[37], the Ni₃N/Ni/NF electrode was synthesized from the electrodeposition of Ni particles on a nickel foam followed by thermal nitridation. A piece of clean nickel foam

(0.5 cm × 0.5 cm) was used as the working electrode. A carbon rod was used as the counter electrode. The Ni/NF was synthesized through electrodeposition at a constant current density of −1.0 A cm$^{-2}$ for 500 s in a two-electrode cell containing NiCl$_2$ (0.1 M) and NH$_4$Cl (2.0 M) under Ar without stirring. The Ni$_3$N/Ni/NF was obtained from the thermal annealing of Ni/NF under an NH$_3$ flow at 300 °C for 6 h with a ramping rate of 10 °C min$^{-1}$.

## Physical characterization

The morphology features of samples were assessed by using scanning electron microscopy (FEI XL30, 15 kV), as well as transmission electron microscopy (TEM), selective area electron diffraction (SAED, 40 μm aperture), and scanning transmission electron microscopy (STEM) with an FEI Tecnai Osiris (200 kV). Samples for TEM were prepared by suspending dry Cu$_3$Ag$_7$/CF electrocatalyst on carbon-coated 200-mesh copper TEM grids (Ted Pella 01894-F). STEM−energy-dispersive X-ray spectroscopy (EDX) maps were collected using Bruker Esprit 1.9 software and averaged over 8 scans. X-ray diffraction (XRD) patterns were collected on a Philips X'Pert Pro PW3040/00 (PAN analytical) instrument. The scan range was set from 20° to 90° (in 2$\theta$) with a Cu-tube operated at 45 kV and 40 mA. The evolved H$_2$ through the electrolysis process was quantified by gas chromatography (GC, SRI 8610C) equipped with a Molecular Sieve 13 packed column, a HayesSep D packed column, and a thermal conductivity detector, and Ar was used as the carrier gas. The Cu and Ag quantities of the sample were analyzed via inductively coupled plasma mass spectrometry (ICP-MS, Agilent 7700 series) in 2% nitric acid. $^1$H NMR spectra were recorded in the designated solvents on a Bruker AV 400 MHz spectrometer. UV−vis absorption spectra were collected on an Agilent 8454 UV−Vis Spectrophotometer.

## Electrochemical measurements

All catalysts on RDE were tested on an electrochemical workstation (VMP-3 potentiostat, Biologic Science Instrument) with a modulated speed rotator (PINE Research Instrumentation) at room temperature in a standard three-electrode configuration. An RDE (0.196 cm$^2$) was utilized as the working electrode, while a Pt mesh and a Hg/HgO (1.0 M KOH) electrode were used as the counter electrode and reference electrode, respectively. All potentials herein were referenced to the reversible hydrogen electrode (RHE) through calibration via Pt foil as the working electrode in the H$_2$-saturated electrolyte. All potentials measured in 1.0 M KOH were converted to the value versus RHE according to the equation: $E_{RHE} = E_{Hg/HgO} + 0.924$ V. Commercial formaldehyde (37 wt%) solution was directly used as the oxidation substrate. The Cu/RDE, Ag/RDE, and Cu$_x$Ag$_{10-x}$/RDE were treated using cyclic voltammetry (CV) from 0 to 0.3 V$_{RHE}$ at 50 mV s$^{-1}$ for 20 cycles to obtain metallic Cu or Ag in 1.0 M KOH under Ar prior to the HCHO electrooxidation (FOR) test (Supplementary Fig. 20). All CVs of FOR were performed in the electrolyte under Ar at a scan rate of 10 mV s$^{-1}$ and a rotation rate of 1500 rpm without iR correction in a standard three-electrode configuration. For investigating the effect of [OH$^-$], 1.0 M anion concentration anolyte consisted of NaOH and NaClO$_4$ with different concentrations to avoid the interference of conductivity change. All potentials herein were referenced to the reversible hydrogen electrode (RHE) through calibration using Pt foil as the working electrode in H$_2$-saturated electrolytes with different OH$^-$ concentrations. The electrochemical double-layer capacitance ($C_{dl}$) measurements were performed using cyclic voltammetry, which was collected in a non-Faradaic region with various scan rates ranging from 20 to 200 mV s$^{-1}$ at potentials between 0.07 and 0.17 V vs. RHE in 1.0 M KOH under Ar. For paraformaldehyde oxidation on RDE, all experimental procedures were the same as FOR.

The measurements of FOR on Cu$_3$Ag$_7$/CF, HER on Ni$_3$Ni/Ni/NF, and OER on Ni/NF were conducted on the VMP-3 potentiostat in an H-cell with an anion exchange membrane (Fumasep FAA-3-50), using a Pt mesh as the counter electrode and a Hg/HgO (1.0 M KOH) as the reference electrode, respectively. Cu$_3$Ag$_7$/CF was treated using cyclic voltammetry from 0 to 0.3 V$_{RHE}$ at 50 mV s$^{-1}$ for 20 cycles to obtain metallic Cu or Ag in 1.0 M KOH under Ar prior to the HCHO electrooxidation test. All CVs (FOR, HER and OER) in a three-electrode configuration were collected at a scan rate of 10 mV s$^{-1}$ without iR correction. The two-electrode electrolysis was performed on an electrochemical workstation in H-cell with an anion exchange membrane. For conventional water electrolysis, the Ni/NF and Ni$_3$Ni/Ni/NF were employed as the anode and cathode in 1.0 M KOH, respectively. For a two-electrode electrolyzer of FOR/HER, Cu$_3$Ag$_7$/CF was the anode and Ni$_3$N/Ni/NF was the cathode, in which 1.0 M KOH and 0.6 M HCHO were used as the anolyte and 1.0 M KOH as the catholyte. The uncompensated resistance (Ru) of 1.0 M KOH in the absence or presence of HCHO was determined by the Current Interrupt (CI) method of a VMP-3 potentiostat and the value of Ru was measured as 15 ± 0.2 Ω. All CVs in a two-electrode configuration were collected under Ar at a scan rate of 10 mV s$^{-1}$ with iR compensation by the automatic CI method with a value of 90% × Ru through the EC-lab software. For the gas products analysis from the anode chamber, different cell voltages (0.2–0.8 V) were applied for chronoamperometry where Cu$_3$Ag$_7$/CF was employed as the anode and Ni$_3$N/Ni/NF as the cathode in a two-electrode configuration. The Faradaic efficiency analysis of H$_2$ and formate production in the cathode and anode chambers was performed from five consecutive 1 h controlled-current electrolysis (150 mA) using the Cu$_3$Ag$_7$/CF and Ni$_3$N/Ni/NF couple but fresh electrolyte for each cycle. The chronopotentiometry test was performed by five consecutive controlled-current electrolysis conducted at 100 and 500 mA/cm$^2$ in a fresh anolyte (1.0 M KOH and 0.6 M HCHO) of each cycle using the Ni$_3$N/Ni/NF(−)||Cu$_3$Ag$_7$/CF(+) electrode couple. The chronoamperometry measurement was carried out at a cell voltage of 0.6 V with the periodic replenishment of fresh HCHO back to its original 0.1 M concentration. For the electrocatalytic paraformaldehyde oxidation on Cu$_3$Ag$_7$/CF, all experimental procedures were the same as FOR. The long-term stability test was carried out in a home-made flow cell with serpentine flow channels (1 cm × 1 cm) using Cu$_3$Ag$_7$/CF as the anode and Ni$_3$N/Ni/NF as the cathode on a Biologic VMP-3 potentiostat without iR-compensation. The catholyte (1.0 M KOH) and anolyte (1.0 M KOH and 10.0 g/L PFA) were fed into the cell at a flow rate of 50 mL min$^{-1}$ by a peristaltic pump (Peri-Star Pro Peristaltic Pump, World Precision Instruments) and recycled. The electrolyte was refreshed every 24 h. An anion exchange membrane (Fumasep FAA-3-50) was used to separate the anolyte and catholyte.

## Product analysis

The evolved H$_2$ in both anode chamber and cathode chamber of a two-electrode electrolyzer of FOR/HER was analyzed by gas chromatography (GC, SRI 8610C) equipped with a Molecular Sieve 13 packed column, a HayesSep D packed column, and a thermal conductivity detector. The oven is kept at 80 °C using Ar as carrier gas. The quantity of H$_2$ production was determined via a water displacement method.

The concentration of formaldehyde during electrolysis was quantified via UV−vis absorption measurement following the Hantzsch reaction[39]. Ammonium acetate (15.4 g) in water (50 mL), glacial acetic acid (0.3 mL), and acetylacetone (0.2 mL) were mixed to form a solution under stirring, which was further diluted with water (49.5 mL). To measure the concentration of formaldehyde, 20.0 μL of anolyte was acidified by 20 μL 2.0 M HCl and then diluted 2500 times prior to and post-electrolysis. Subsequently, 2.0 mL of the diluted solution was mixed with the acetylacetone solution (2.0 mL), which was further heated to 60 °C for 10 min. After cooling for 10 min, the absorbance of the sample solution at 413 nm was measured. The quantification of HCHO was obtained from the calibration curves by applying standard solutions with known concentrations of commercially purchased pure HCHO.

The identification and quantification of formic acid and methanol were determined from the $^1$H NMR using calibration curves with t-butanol (10.0 mM) as an internal standard. 100 μL electrolyte from anolyte and catholyte prior to and post electrolysis was acidified by 20 μL HCl (37%) and then added into 500 μL D$_2$O. $^1$H NMR was recorded using a water suppression method.

The Faradaic efficiency was calculated on the basis of the following equation:

$$FE\,(\%) = (nF \times N / Q_{\text{total charge passed}}) \times 100 \qquad (1)$$

where $n$ is the number of electrons transferred for each product molecule, $F$ is Faraday's constant (96,485 C mol$^{-1}$), $N$ is the mole number of products and $Q$ is the total passed charge.

The carbon balance (%) of the electrooxidation process was calculated using the following equation:

$$\text{carbon balance}\,(\%) = \frac{\text{mol of organic products}}{\text{mol of formaldehyde consumed}} \times 100 \qquad (2)$$

## Theoretical computation

Density functional theory (DFT) calculations were performed by using Vienna Ab initio Simulation Package (VASP)[44,45]. The Perdew–Burke–Ernzerhof (PBE) functional within the generalized gradient approximation (GGA) was used for electron exchange correlation[46]. Projector augmented wave (PAW) potential was used to treat the nuclei-electron interaction[47,48]. An energy cutoff of 400 eV was chosen for plane wave basis sets. The atomic positions were relaxed until the force on each atom was <0.05 eV/Å. Electronic energies converged within 10$^{-4}$ eV. Transition states were searched using the Climbing Image Nudged Elastic Band approach[49,50]. Frequency analysis was carried out to ensure that there was only a single imaginary frequency for the transition state.

The bulk model of Cu$_3$Ag$_7$ was built based on the bulk model of Ag. Starting with a 2 × 2 × 2 supercell of bulk Ag with 16 Ag atoms, we replaced two Ag atoms with Cu and obtained four different geometries for Cu$_2$Ag$_{14}$. After optimization, the lowest energy structure was used as a starting structure to generate six different geometries for Cu$_3$Ag$_{13}$ from replacing one Ag in Cu$_2$Ag$_{14}$ with Cu. Repeating the optimization and the single-atom substitution of the lowest-energy structure, seven geometries of Cu$_4$Ag$_{12}$ and two geometries of Cu$_5$Ag$_{11}$ were tested and compared. In total, over 20 bulk geometries were examined for a stable structure for Cu$_5$Ag$_{11}$ (Cu/Ag = 0.45) which was used to approximate the bulk Cu$_3$Ag$_7$ (Cu/Ag = 0.43) composition. A slab with five layers in a (3 × 3) lateral supercell was built for Cu$_3$Ag$_7$, Ag, and Cu (111) surfaces and sampled by 2 × 2 × 1 $k$-point mesh. The bottom two layers were fixed, and the other atoms were all relaxed during the structural optimization. The vacuum layer was set to be 15 Å. Coordinates for the bulk model and the (111) surface of the bimetallic system are provided in Supplementary Data 1.

## Data availability

All data are available in the manuscript, the supplementary materials and from the authors on request. Source data are provided as a Source Data file. Source data are provided with this paper.

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

## Acknowledgements

Y.S. acknowledges the support of the National Science Foundation (CHE-1914546 and CHE-2102220) and Herman Frasch Foundation (820-HF17). DFT computation (D.J.) was sponsored by the National Science Foundation (CHE-2102191).

## Author contributions

Y.S. conceived and designed this research. G.L. and G.H. synthesized and characterized the catalysts and performed electrocatalysis experiments. L.W. and D.J. conducted the theoretical computation. G.L. and X.C. carried out the ICP-MS measurements. N.K.M. and P.R.K. performed TEM and HR-TEM measurements. All authors contributed to the analysis and interpretation of the results. G.L., Y.S., and D.J. wrote the manuscript.

## Competing interests

Y.S. has filed a provisional patent application related to this manuscript (US patent provisional 63/440,048). All other authors declare no competing interests.
