## [Peer Review File · Nature Communications]

REVIEWER COMMENTS

Reviewer #1 (Remarks to the Author):

In this work, the authors synthesized a Cu₃Ag₇ electrocatalyst to efficiently produce H₂ from formaldehyde. They introduced a two-electrode electrolyzer employing Cu₃Ag₇ and Ni₃N/Ni as the anode and cathode electrocatalysts, respectively. The electrolyzer was able to produce H₂ at both anode and cathode with a 200% faradaic efficiency, reaching a current density of 500 mA/cm² at a cell voltage of 0.60 V. The results of this work are interesting. However, there are some issues to be clarified before publication. The details are listed below.

- 1) There are some reports on the coupling of formaldehyde oxidation and water reduction to produce H₂ at both anode and cathode. It would be better to refer these papers in the introduction section.
- 2) Why did the authors choose Ag and Cu as catalytic materials? Furthermore, the authors should discuss in detail why the Cu₃Ag₇ electrocatalyst has selectivity for HCHO, not methanol and formate.
- 3) The authors mentioned that experimental characterizations showed that the (111) facets are preferentially exposed for the catalysts, and they constructed (111) surfaces as calculation models. However, detailed analysis showing that the (111) facets are mainly exposed on the surfaces of the catalysts is not provided. The facet exposed on the surface cannot be exactly indexed by the XRD pattern.

Reviewer #2 (Remarks to the Author):

The manuscript reported by Li et al. presents an alternative process to H₂ generation with high efficiency using Cu₃Ag₇ electrocatalysts. The method is relevant to the area, and the methodology used is apparently sound, but I am not a specialist in the experimental details and will not make any consideration of that. The text is very well written; the introduction section adequately contexts the problem and cites pertinent references.

Concerning the computational aspects, the work has some deficiencies that should be considered.

Authors justify from experimental evidence that the (111) surfaces have a higher reaction activity, and they present one model for each system. The adsorption energies for the $\text{H}_2\text{C}(\text{OH})\text{O}^*$ intermediate show that Cu_3Ag_7 is the most favored.

However, no details about the modeling of this system are found in the text or the supporting information.

1. It is said that 'various bulk configurations were explored to reach a low-energy structure'. Then, a) how many; b) how was the building process of those bulk configurations?
2. When creating the slab model oriented in the (111) direction from the minimal energy bulk structure, different terminations can be exposed, and different chemical environments are present. Is expected that all of them are represented by the site reported in the manuscript? A more extensive exploration of other adsorption sites should be done.
3. The adsorption energies reported are based on the calculated total DFT energies. It should be more appropriate to compare the free energies.
4. What is the role of the binary metal surface in the stabilization of the intermediate? A discussion on the electronic structure of the systems should be included.

Reviewer #3 (Remarks to the Author):

The manuscript "H₂ Production of 200% Faradaic Efficiency from Electrocatalytic Water Reduction Coupled with Formaldehyde Oxidation via a Copper-Silver Electrocatalyst" by G. Li, G. Han, L. Wang, X. Cui, D.e. Jiang and Y. Sun report a series of $\text{Cu}_x\text{Ag}_{1-x}$ catalysts for formaldehyde oxidation as anode reaction in water electrolysis alternative to the oxygen evolution reaction, demonstrating that the catalyst achieves high currents at substantially low electrode potentials. Moreover, coupled with a Ni-based catalyst as cathode, the electrochemical system displays full FE efficiency for the two half-reactions, demonstrating production of H₂ in both the anode and cathode compartments. The work is well structured, very interesting and highly relevant for the field. Yet, there are few remarks that still need to be addressed before its publication:

Major remarks:

1. Authors should provide evidence of the reproducibility of the synthesis method and evaluation methods, for instance by showing a set of measurements (voltammetry, chronometry, and product analysis and quantification) for three independent electrodes.

2. In page 6, authors state “Increasing the HCHO concentration beyond 0.6 M resulted in decreased anodic current, likely due to more disproportionation of HCHO at higher concentration”. What exactly controls the disproportionation, is it the concentration of HCHO or its concentration relative to that of OH⁻? In other words, if one would have a more concentrated electrolyte, can one also vary the HCHO concentration while keeping disproportionation low? It would be useful for potential readers to include this in the discussion, particularly considering that industrial alkaline electrolyzers operate with highly concentrated KOH solutions. Detection of methanol and formate at higher OH⁻ concentrations without application of potentials could be helpful to clarify this.

Minor remarks:

3. I struggle with the term “200% FE”; since the concept of FE is typically associated with a particular catalyst for a particular half-reaction, it is difficult to grasp from the title what is meant there. After reading the manuscript, it is clear that two different catalyst display 100% FE for two different reactions, which happen to deliver the same (desired) product. Yet, my impression is that summing the FEs up is rather misleading (because these are two different reactions and two different catalysts). I suggest the authors to reconsider the term “200% FE” and find a more suitable way to describe the high efficiency of the presented system.

4. Page 4, about electrodeposition, authors indicate twice (lines 92 and 106) to see the Supporting Information for details on the electrodeposition procedure, however this information is not featured in the SI.

5. Fig. 1, I suggest to plot the three catalysts together or alternatively to stack the two plots to facilitate the comparison in terms of electrode potential.

6. Page 4, “more positive onset potential (0.2 V vs RHE)” how is the onset potential determined? This should be clearly defined in this part of the discussion.

7. In the experimental, authors indicate “The Cu/RDE, Ag/RDE, and Cu_xAg_{10-x}/RDE were treated using cyclic voltammetry (CV) from 0 VRHE to 0.3 VRHE at 50 mV s⁻¹ for 20 cycles to obtain metallic Cu or Ag in 1.0 M KOH under Ar prior to the HCHO electrooxidation (FOR)”. Please show at least one set of CVs for each catalyst e.g. in the supporting information.

8. In the experimental, authors indicate “All CVs in a two-electrode configuration were collected under Ar at a scan rate of 10 mV s⁻¹ with 90% iR correction” How was the uncompensated resistance determined? It would be interesting to also provide information of the uncompensated resistance for the different electrolyte compositions used.

Thanks for the interesting manuscript.

Reviewer #4 (Remarks to the Author):

This manuscript demonstrates the production of hydrogen at both the anode and cathode of an electrochemical cell by coupling formaldehyde oxidation with hydrogen evolution, thus resulting in a reduced cell potential compared to water splitting. The scope of this manuscript is well aligned with Nature Communications, given the focus on an emerging system for efficient hydrogen production. Overall, the manuscript is thorough and well-written. The current densities achieved are significant and the high overall faradaic efficiency to hydrogen is notable. It should be mentioned, though, that coupling of organic molecule oxidation at the anode with hydrogen evolution at the cathode has been demonstrated previously as noted by the authors. Thus, the concept is not novel, but the demonstrated performance metrics represent a significant achievement. My major concern with the manuscript is around the interpretation and discussion/consideration of durability. Please see my specific comments below that I believe need to be addressed prior to further consideration:

1. If run in a continuous process under less dilute conditions, how does the pH of the electrolyte change over time in the anode and cathode chambers given the OH⁻ balance? How would this affect the durability of the process, especially considering moving beyond an H-cell into a scale-up electrochemical device?
2. The authors utilize Fig. 3E and 3F as evidence for the stability of the process and the electrocatalyst. However, in Fig. 3E, the slope of the voltage vs. time data at 500mA/cm² is increasing quite considerably with each cycle (even with full replacement of the electrolyte). Further, in Fig. 3F, the maximum current density obtained decreases for each subsequent cycle. In both cases, this data is only collected over less than 10 hours, which is a short period of time relative to industrial processes. The authors need to discuss these results in detail and include interpretation about what it means for the long-term durability of this process, rather than just stating the cycles are “nearly identical”.

Point-by-Point Response Letter

Reviewer #1:

In this work, the authors synthesized a Cu₃Ag₇ electrocatalyst to efficiently produce H₂ from formaldehyde. They introduced a two-electrode electrolyzer employing Cu₃Ag₇ and Ni₃N/Ni as the anode and cathode electrocatalysts, respectively. The electrolyzer was able to produce H₂ at both anode and cathode with a 200% faradaic efficiency, reaching a current density of 500 mA/cm² at a cell voltage of 0.60 V. The results of this work are interesting. However, there are some issues to be clarified before publication. The details are listed below.

1) There are some reports on the coupling of formaldehyde oxidation and water reduction to produce H₂ at both anode and cathode. It would be better to refer these papers in the introduction section.

Response: Thanks for the reviewer's comments. After a thorough literature search, we noticed that formaldehyde has been used as a feedstock for electrochemical oxidation to lower the input voltage of H₂ production at the cathode (*ACS Appl. Mater. Interfaces* **2021**, *13*, 38256-38265.; *Chem. Eng. J.* **2021**, *426*, 129214.; *ACS Sustainable Chem. Eng.* **2022**, *10*, 7108-7116.; *J. Alloys Compd.* **2022**, *925*, 166748.). However, in these works, H₂ was only produced at the cathode. Although there was a report on formaldehyde oxidation to produce H₂ at the anode (*Electrochim. Acta* **2020**, *333*, 135542), the cathodic reaction was O₂ reduction, not H₂ production. In fact, several biomass-derived aldehydes (e.g., furfural, 5-hydroxymethyl furfural) have been utilized for low-potential oxidation to release H₂ at the anode; while in these systems, cathodic H₂ production was also reported (*Nat. Catal.* **2022**, *5*, 66-73; *Energy Environ. Sci.* **2022**, *15*, 4175-4189). Overall, to the best of our knowledge, there has not been any report on the coupling of formaldehyde oxidation and water reduction to produce H₂ at **both** anode and cathode simultaneously.

2) Why did the authors choose Ag and Cu as catalytic materials? Furthermore, the authors should discuss in detail why the Cu₃Ag₇ electrocatalyst has selectivity for HCHO, not methanol and formate.

Response: Formaldehyde (HCHO) has been extensively investigated as a liquid organic hydrogen storage molecule for H₂ production under thermocatalytic conditions. During the dehydrogenation process of HCHO, the most critical step is the breaking of its C-H bond. Several metal catalysts, such as Cu, Ag, Au, Pd, and Pt, have shown excellent performance for thermocatalytic HCHO oxidation to produce H₂ in alkaline media (*Int. J. Hydrogen Energy* **2008**, *33*, 2225-2232; *Int. J. Hydrogen Energy* **2010**, *35*, 7177-7182; *Nano Energy* **2014**, *8*, 103-109; *Int. J. Hydrogen Energy* **2014**, *39*, 9114-9120; *Int. J. Hydrogen Energy* **2015**, *40*, 1752-1759; *Inorg. Chem. Front.*, **2017**, *4*, 1704-1713; *Catal. Sci. Technol.*, **2019**, *9*, 5292-5300; *Appl. Surf. Sci.* **2022**, *589*, 152908). Different from thermocatalytic HCHO dehydrogenation, electrocatalytic HCHO oxidation can proceed via either the one-electron transfer (HCHO + 2OH⁻ → HCOO⁻ + 1/2H₂ + H₂O + e⁻) or two-electron transfer (HCHO + 3OH⁻ → HCOO⁻ + 2H₂O + 2e⁻) processes (*J. Appl. Electrochem.* **1981**, *11*, 387-393). Only the one-electron transfer process will lead to H₂ production, while the two-electron transfer process involves hydrogenation oxidation, which is a competing anodic reaction and should be avoided. Since Pt and Pd are well-known electrocatalysts for hydrogen oxidation, they were not selected. In addition, because of its high cost, Au was not employed as well. Since hydrogen oxidation is not favorable on Cu and

Ag (*ACS Catal.* **2018**, *8*, 6665-6690; *ACS Nano* **2022**, *16*, 5153-5183), therefore we chose Cu and Ag as model electrocatalysts for the low-potential oxidation of HCHO to produce H₂ at the anode.

To address the reviewer's 2nd question, we collected the CV curves of Cu₃Ag₇/RDE in the region of 0 to 0.4 V vs RHE in 1.0 M KOH after the addition of 0.1 M HCOOH, 0.1 M CH₃OH, and 0.1 M HCHO, respectively. As shown in **Figure R1**, the increase in anodic current on Cu₃Ag₇/RDE was negligible upon the addition of either HCOOH or CH₃OH. However, a rapid anodic current rise was observed once HCHO was added, indicating that Cu₃Ag₇/RDE could effectively catalyze the electrooxidation of HCHO but not HCOOH or CH₃OH within this potential region. This could be due to the difference in the dissociation energy of their C-H bonds. Among the C-H bonds in HCHO, HCOOH, and CH₃OH at 298 K, the bond dissociation energy of C-H bond in H-CHO ($88.0 \pm 0.2 \text{ kcal mol}^{-1}$) is lower than those in H-CH₂OH ($96.1 \pm 0.2 \text{ kcal mol}^{-1}$) and H-COOH ($\geq 96 \pm 1 \text{ kcal mol}^{-1}$), and it is also the lowest compared to other bonds (C-O and C=O) (*Acc. Chem. Res.* **2003**, *36*, 255-263). Even though HCOOH and CH₃OH could be oxidized on copper/silver oxides, they require much more positive potential (*J. Electroanal. Chem.* **2000**, *495*, 71-78.; *Electrochim. Acta* **2004**, *49*, 4999-5006), way beyond 0.4 V vs RHE. We have added relevant discussion in the revised manuscript (highlight text on Page 10-11 in the revised manuscript).

Figure R1. CV curves of Cu₃Ag₇/RDE in 1.0 M KOH upon the addition of 0.1 M HCOOH (black), 0.1 M CH₃OH (blue), and 0.1 M HCHO (red) collected at 1500 rpm and 10 mV/s under Ar.

3) The authors mentioned that experimental characterizations showed that the (111) facets are preferentially exposed for the catalysts, and they constructed (111) surfaces as calculation models. However, detailed analysis showing that the (111) facets are mainly exposed on the surfaces of the catalysts is not provided. The facet exposed on the surface cannot be exactly indexed by the XRD pattern.

Response: We appreciate the valuable comment from the reviewer. To further investigate the structure of Cu₃Ag₇/CF, high-resolution transmission electron microscopy (HRTEM) combined with selected area electron diffraction (SAED) were performed. **Figure R2a** presents a clear dendritic structure in line with the SEM images (**Fig. 3a** and **Fig. S5c**). The HRTEM images

(Figure R2b-2c) show the well-resolved lattice fringes with an inter-planar distance of 0.237 nm and 0.208 nm corresponding to the (111) crystal plane of cubic Ag and Cu, respectively. The distinct diffraction rings from SAED (Figure R2d) demonstrate the polycrystalline nature and can be indexed into the (111), (200), (220), and (311) planes of Ag and Cu, respectively, in good agreement with the XRD results. High-angle-annular dark-field STEM (HAADF-TEM) and energy-dispersive X-ray spectroscopy (EDX) element mapping images reveal a uniform distribution of Cu and Ag throughout the dendrites (Figure R2e). Collectively, we conclude that the (111) facets are mainly exposed on the surfaces of our electrocatalysts. Relevant discussion has been added in the revised manuscript (highlighted text on Page 7-8 in the revised main text file).

Figure R2. Structural and compositional analysis of $\text{Cu}_3\text{Ag}_7/\text{CF}$. (a) TEM, (b-c) HR-TEM, (d) SAED, and (e) HAADF and EDX mapping images of $\text{Cu}_3\text{Ag}_7/\text{CF}$.

Reviewer #2:

The manuscript reported by Li et al. presents an alternative process to H₂ generation with high efficiency using Cu₃Ag₇ electrocatalysts. The method is relevant to the area, and the methodology used is apparently sound, but I am not a specialist in the experimental details and will not make any consideration of that. The text is very well written; the introduction section adequately contexts the problem and cites pertinent references.

Concerning the computational aspects, the work has some deficiencies that should be considered. Authors justify from experimental evidence that the (111) surfaces have a higher reaction activity, and they present one model for each system. The adsorption energies for the H₂C(OH)O* intermediate show that Cu₃Ag₇ is the most favored.

However, no details about the modeling of this system are found in the text or the supporting information.

1. It is said that ‘various bulk configurations were explored to reach a low-energy structure’. Then, a) how many; b) how was the building process of those bulk configurations?

Response: As the reviewer suggested, we have added the following details in the Methods section of the text: “Starting with a 2×2×2 supercell of bulk Ag with 16 Ag atoms, we replaced two Ag atoms with Cu and obtained four different geometries for Cu₂Ag₁₄. After optimization, the lowest energy structure was used as a starting structure to generate six different geometries for Cu₃Ag₁₃ from replacing one Ag in Cu₂Ag₁₄ with Cu. Repeating the optimization and the single-atom substitution of the lowest-energy structure, seven geometries of Cu₄Ag₁₂ and two geometries of Cu₅Ag₁₁ were tested and compared. In total, over 20 bulk geometries were examined to a stable structure for Cu₅Ag₁₁ which was used to approximate the bulk Cu₃Ag₇ composition.”

2. When creating the slab model oriented in the (111) direction from the minimal energy bulk structure, different terminations can be exposed, and different chemical environments are present. Is expected that all of them are represented by the site reported in the manuscript? A more extensive exploration of other adsorption sites should be done.

Response: When the bulk model was constructed from the process described above, we found that most of the Cu atoms are in a (100) plane. This led to a rather simple surface termination for the (111) surface, where (100) and (111) planes intersect at a line of Cu atoms on the surface. Coordinates for the bulk model and the (111) surface of the bimetallic system are provided in the Supplementary Information. We have indeed explored various adsorption sites of the H₂C(OH)O* intermediate on the model (111) surface and the one in **Fig. 5** (see the main text) is the most stable one.

3. The adsorption energies reported are based on the calculated total DFT energies. It should be more appropriate to compare the free energies.

Response: We agree that free energies would be ideal; however, to simulate the real experimental conditions, many factors complicate the free-energy evaluations and introduce uncertainties, including the solvation environment, the local pH, etc. Because our focus is on the relative comparison of Cu, Ag, and Cu₃Ag₇ surfaces, we feel that the relative adsorption energy is a reasonable approximation to the relative free-energy change, considering the uncertainties of the free-energy evaluations and the benefit of error cancellation when focusing on the relative trend.

4. What is the role of the binary metal surface in the stabilization of the intermediate? A discussion on the electronic structure of the systems should be included.

Response: We thank the reviewer for the great suggestion. It can be seen from the adsorption geometry of $\text{H}_2\text{C}(\text{OH})\text{O}^*$ on the bimetallic surface (**Fig. 5C** and **Figure R3** inset) that the alkoxy O atom of $\text{H}_2\text{C}(\text{OH})\text{O}^*$ interacts strongly with Cu and Ag atoms. We analyzed the site-projected, orbital-resolved electronic density of states of the adsorption geometry. As it can be seen from **Figure R3**, there are strong orbital mixings of O 2p states with Cu 3d states at -1.5 to -1.0 eV as well as with Ag 4d states at -6 to -5 eV and -4 to -3 eV. Apparently, the existence of two d-bands on the bimetallic surface provides more flexibility in adsorbing and stabilizing the $\text{H}_2\text{C}(\text{OH})\text{O}^*$ intermediate. We think that this is an important insight, which we have included in the text together with **Figure R3** (see **Supplementary Fig. S16c**).

Figure R3. Projected density of states of the O atom of $\text{H}_2\text{C}(\text{OH})\text{O}^*$ and the Cu and Ag atoms at the adsorption site (see inset for the atoms being analyzed; +/- y-values indicate spin up/down channels).

Reviewer #3:

The manuscript “H₂ Production of 200% Faradaic Efficiency from Electrocatalytic Water Reduction Coupled with Formaldehyde Oxidation via a Copper-Silver Electrocatalyst” by G. Li, G. Han, L. Wang, X. Cui, D.e. Jiang and Y. Sun report a series of Cu_xAg_{1-x} catalysts for formaldehyde oxidation as anode reaction in water electrolysis alternative to the oxygen evolution reaction, demonstrating that the catalyst achieves high currents at substantially low electrode potentials. Moreover, coupled with a Ni-based catalyst as cathode, the electrochemical system displays full FE efficiency for the two half-reactions, demonstrating production of H₂ in both the anode and cathode compartments. The work is well structured, very interesting and highly relevant for the field. Yet, there are few remarks that still need to be addressed before its publication:

Major remarks:

1. Authors should provide evidence of the reproducibility of the synthesis method and evaluation methods, for instance by showing a set of measurements (voltammetry, chronometry, and product analysis and quantification) for three independent electrodes.

Response: Following the reviewer’s suggestions, three electrodes (Cu₃Ag₇/CF-1, Cu₃Ag₇/CF-2, and Cu₃Ag₇/CF-3) were prepared following the same method as described in the manuscript and then evaluated to confirm the reproducibility.

Cyclic voltammetry and chronoamperometry measurements of HCHO oxidation using the three electrodes were performed using the same H-cell in a three-electrode configuration. As shown in **Figure R4a**, the CV curves of all the electrodes collected nearly overlap with other each from 0 to 0.40 V vs RHE. A 5 h chronoamperometry experiment was performed for each electrode at 0.3 V_{RHE} and the corresponding I-t curves were very similar (**Figure R4b**).

For the Faradaic efficiency analysis and quantifications of products, controlled-current (150 mA) electrolysis for 1 h was performed for each electrode. Comparing the experimentally measured amount of H₂ with the calculated amount of H₂ based on passed charge during each electrolysis on three independent electrodes confirmed that 100% Faradaic efficiency was achieved for anodic H₂ production on each electrode (**Figure R5** and **Table R1**). The organic products in the liquid phase of the anode and cathode chamber were determined and quantified via ¹H NMR using *t*-butanol as an internal standard (**Figure R6**). 100% Faradaic efficiency of formate production from electrocatalytic oxidation of HCHO was also achieved for each electrode (**Figure R5d**).

In summary, these above results demonstrate the great reproducibility of our synthesis method for the preparation of desirable electrocatalysts.

Figure R4. (a) CV curves of Cu₃Ag₇/CF-1, Cu₃Ag₇/CF-2, and Cu₃Ag₇/CF-3 in 1.0 M KOH with 0.6 M HCHO collected at 10 mV/s. (b) Chronoamperometric curves of Cu₃Ag₇/CF-1, Cu₃Ag₇/CF-2, and Cu₃Ag₇/CF-3 at 0.3 V_{RHE} in 1.0 M KOH with 0.6 M HCHO.

Figure R5. (a-c) Comparison of the experimentally measured H₂ amounts with the theoretical H₂ amounts along the passed charge for the anode chambers. (d) Faradaic efficiencies of H₂ and formate production. Condition: The electrolysis experiments were conducted at 150 mA for 1 h in 1.0 M KOH with 0.6 M HCHO using the same H-cell in a three-electrode configuration. Cu₃Ag₇/CF-1, Cu₃Ag₇/CF-2, and Cu₃Ag₇/CF-3 were used as the working electrodes individually.

Table R1. The Faradaic efficiency (FE) of anodic products after each electrolysis with different working electrodes: Cu₃Ag₇/CF-1, Cu₃Ag₇/CF-2, and Cu₃Ag₇/CF-3.

Electrodes	n _{HCOOH} (mmol)		n _{CH₃OH} (mmol)		n _{HCOO⁻} (mmol) (Calculated)	FE _{HCOO⁻}	n _{H₂} (mmol)	FE _{H₂}
	From HCHO oxidation	From Cannizzaro reaction	From Cannizzaro reaction	From Cannizzaro reaction				
Cu ₃ Ag ₇ /CF-1	5.59	2.63	2.63	2.63	(540C) 5.60	99.8%	2.81	100%
Cu ₃ Ag ₇ /CF-2	5.63	2.76	2.76	2.76	(540C) 5.60	100.5%	2.81	100%
Cu ₃ Ag ₇ /CF-3	5.57	2.65	2.65	2.65	(540C) 5.60	99.5%	2.81	100%

Figure R6. The ¹H NMR (D₂O, 400 MHz) of organic products in the liquid phase of the anode and cathode chamber during each electrolysis with different electrodes (Cu₃Ag₇/CF-1, Cu₃Ag₇/CF-2, and Cu₃Ag₇/CF-3) at 150 mA for 1 h using the same H-cell in a three-electrode configuration, in which 1.0 M KOH was used as the catholyte and 1.0 M KOH with 0.6 M HCHO as the anolyte.

2. In page 6, authors state “Increasing the HCHO concentration beyond 0.6 M resulted in decreased anodic current, likely due to more disproportionation of HCHO at higher concentration”. What exactly controls the disproportionation, is it the concentration of HCHO or its concentration relative to that of OH⁻? In other words, if one would have a more concentrated electrolyte, can one also vary the HCHO concentration while keeping disproportionation low? It would be useful for potential readers to include this in the discussion, particularly considering that industrial alkaline electrolyzers operate with highly concentrated KOH solutions. Detection of methanol and formate at higher OH⁻ concentrations without application of potentials could be helpful to clarify this.

Response: We appreciate the valuable comments from the reviewer. Following the reviewer’s suggestion, formate and methanol were first detected under the electrolysis condition using different anolyte containing different concentrations of HCHO or OH⁻. The controlled-current electrolysis was performed at 150 mA for 1 h in a three-electrode configuration using Cu₃Ag₇/CF as the working electrode. The formate and methanol products in the liquid phase of the anode and cathode chamber were determined and quantified via ¹H NMR using 10.0 mM *t*-butanol as an internal standard. Besides the quantity of formate resulting from the HCHO electrooxidation with 100% FE, all the additional formate was produced from the Cannizzaro reaction (**Table R2**).

When the OH⁻ concentration was maintained at 1.0 M while the initial HCHO concentration increased from 0.6 M to 1.2 M and 3.0 M in the anolyte, formate and methanol obtained from the Cannizzaro reaction increased substantially (**Table R2** and **Figures R7-R8**). The formate produced by disproportionation was 2.63 mmol after 1 h controlled-current electrolysis in 1.0 M KOH with 0.6 M HCHO, while it was 6.69 or 9.28 mmol as HCHO concentration increased by 2 or 5 times.

When the relative ratio of KOH and HCHO concentration remained constant (1/0.6) but the concentrations of both KOH and HCHO increased, the disproportionation became more severe (**Table R2** and **Figure R9**). Formate from the Cannizzaro reaction was 48.79 mmol in 5.0 M KOH with 3.0 M HCHO, which was much higher than that (2.63 mmol) in 1.0 M KOH with 0.6 M HCHO. The results showed that when the ratio between KOH and HCHO was kept constant, an increase in their concentrations would facilitate the disproportionation reaction.

In addition, we also quantified the amount of formate and methanol at different times without applying potentials. We kept the concentration of HCHO at 0.6 M but varied the KOH concentration. As shown in **Figures R10-R13**, the amounts of formate and methanol both increased as the KOH concentration increased. The formate or methanol produced from disproportionation achieved ~250 mM within 3 h in 5.0 M KOH, while it took about 40 h to reach the same amount in 1.0 M KOH. When the KOH concentration was 0.1 M, the disproportionation occurred very slowly. The concentrations of formate and methanol only reached 50 mM in 40 h.

Table R2. The organic products of each electrolysis in different electrolytes at 150 mA for 1 h in a three-electrode H-cell configuration using Cu₃Ag₇/CF as the working electrode.

Electrolyte ^a	n _{HCOOH} (mmol)		n _{CH₃OH} (mmol)	n _{HCOO⁻} (mmol) (Calculated)
	From HCHO oxidation	From Cannizzaro reaction	From Cannizzaro reaction	
Anolyte: 1.0 M KOH 0.6 M HCHO Catholyte: 1.0 M KOH	5.59	2.63	2.63	(540C) 5.60
Anolyte: 1.0 M KOH 1.2 M HCHO Catholyte: 1.0 M KOH	5.64	6.69	6.69	(540C) 5.60
Anolyte: 1.0 M KOH 3.0 M HCHO Catholyte: 1.0 M KOH	5.62	9.28	9.28	(540C) 5.60
Anode: 2.0 M KOH 1.2 M HCHO Catholyte: 2.0 M KOH	5.64	6.80	6.80	(540C) 5.60
Anolyte: 5.0 M KOH 3.0 M HCHO Catholyte: 5.0 M KOH	5.58	48.79	48.79	(540C) 5.60

Note: ^a Both anolyte and catholyte were 50.0 ml.

Figure R7. Formate products from HCHO electrooxidation (dark color) and Cannizzaro reaction (light color) of each electrolysis in different electrolytes at 150 mA for 1 h in a three-electrode H-cell configuration using Cu₃Ag₇/CF as the working electrode.

Figure R8. The ^1H NMR (D_2O , 400 MHz) of organic products in the liquid phase of the anode and cathode chambers after electrolysis using different electrolytes at 150 mA for 1 h in a three-electrode H-cell configuration using $\text{Cu}_3\text{Ag}_7/\text{CF}$ as the working electrode.

Figure R9. The ¹H NMR (D₂O, 400 MHz) of organic products in the liquid phase of the anode and cathode chambers after electrolysis using different electrolytes at 150 mA for 1 h in a three-electrode H-cell configuration using Cu₃Ag₇/CF as the working electrode.

Figure R10. Concentrations of formate (a) and methanol (b) produced from the Cannizzaro reaction with 0.6 M HCHO and different concentrations of KOH without applied potential.

Figure R11. The ^1H NMR (D_2O , 400 MHz) of organic products in 0.1 M KOH in the presence of 0.6 M HCHO for different times without application of potentials using 10.0 mM *t*-butanol as an internal standard.

Figure R12. The ¹H NMR (D₂O, 400 MHz) of organic products in 1.0 M KOH in the presence of 0.6 M HCHO for different times without application of potentials using 10.0 mM *t*-butanol as an internal standard.

Figure R13. The ^1H NMR (D_2O , 400 MHz) of organic products in 5.0 M KOH in the presence of 0.6 M HCHO for different times without application of potentials using 10.0 mM *t*-butanol as an internal standard.

Minor remarks:

3. I struggle with the term “200% FE”; since the concept of FE is typically associated with a particular catalyst for a particular half-reaction, it is difficult to grasp from the title what is meant there. After reading the manuscript, it is clear that two different catalysts display 100% FE for two different reactions, which happen to deliver the same (desired) product. Yet, my impression is that summing the FEs up is rather misleading (because these are two different reactions and two different catalysts). I suggest the authors to reconsider the term “200% FE” and find a more suitable way to describe the high efficiency of the presented system.

Response: In our work, H₂ production with simultaneous 100% anodic FE and 100% cathodic FE exhibited an apparent FE of 200% (one electron leads to two H* and hence one H₂). Following the reviewer’s comments, we have changed the title to “Dual H₂ Production from Electrocatalytic Water Reduction Coupled with Formaldehyde Oxidation via a Copper-Silver Electrocatalyst”. In addition, we also have changed “a 200% Faradaic efficiency” to “an apparent 200% Faradaic efficiency” in the abstract and main text of the revised manuscript.

4. Page 4, about electrodeposition, authors indicate twice (lines 92 and 106) to see the Supporting Information for details on the electrodeposition procedure, however this information is not featured in the SI.

Response: We appreciate the reviewer for pointing out this error. We have changed “see the Supporting Information for details” to “see Methods for details” in the revised manuscript.

5. Fig. 1, I suggest to plot the three catalysts together or alternatively to stack the two plots to facilitate the comparison in terms of electrode potential.

Response: Following the reviewer’s suggestion, we have redrawn the original **Fig.1** as a new **Fig. 2** in the revised manuscript, which is shown as **Figure R14** below.

Figure R14. CV curves of Cu/RDE (blue), Ag/RDE (black), and Cu₃Ag₇/RDE (red) in 1.0 M KOH in the absence (dashed) and presence (solid) of 0.6 M HCHO collected at 1500 rpm and 10 mV/s. Inset shows the expanded CV of copper oxidation on Cu₃Ag₇/RDE.

6. Page 4, “more positive onset potential (0.2 V vs RHE)” how is the onset potential determined? This should be clearly defined in this part of the discussion.

Response: Since onset potential is neither a thermodynamically nor kinetically well-defined parameter, herein we use the potential at a given current density of 0.1 mA/cm² as the onset

potential (following reported literature on O₂ reduction studies). We have added the definition of onset potential in the revised manuscript.

7. In the experimental, authors indicate “The Cu/RDE, Ag/RDE, and Cu_xAg_{10-x}/RDE were treated using cyclic voltammetry (CV) from 0 V_{RHE} to 0.3 V_{RHE} at 50 mV s⁻¹ for 20 cycles to obtain metallic Cu or Ag in 1.0 M KOH under Ar prior to the HCHO electrooxidation (FOR)”. Please show at least one set of CVs for each catalyst e.g. in the supporting information.

Response: Following the reviewer’s comment, we have added the CV curves of Cu/RDE, Ag/RDE, and Cu_xAg_{10-x}/RDE from 0 V_{RHE} to 0.3 V_{RHE} (Figure R14) as Supplementary Fig. 20 in the revised Supplementary Information.

Figure R15. (a-g) CV curves of Cu/RDE, Ag/RDE, and Cu_xAg_{10-x}/RDE from 0 V_{RHE} to 0.3 V_{RHE} at 50 mV s⁻¹ for 20 cycles in 1.0 M KOH under Ar.

8. In the experimental, authors indicate “All CVs in a two-electrode configuration were collected under Ar at a scan rate of 10 mV s^{-1} with 90% iR correction” How was the uncompensated resistance determined? It would be interesting to also provide information of the uncompensated resistance for the different electrolyte compositions used.

Reponse: In our two-electrode H-cell configuration, the uncompensated resistance (R_u) for 1.0 M KOH in the absence or presence of HCHO was determined by the Current Interrupt (CI) method of a VMP-3 potentiostat (Biologic Science Instrument) and the value of R_u was $15 \pm 0.2 \Omega$. The iR compensation was performed by automatic CI method with a value of $90\% \times R_u$ through EC-lab software. We have added the iR compensation information in **Methods** of the revised manuscript.

Thanks for the interesting manuscript.

Reponse: We appreciate the reviewer’s valuable comments in helping us to improve the quality of this work.

Reviewer #4:

This manuscript demonstrates the production of hydrogen at both the anode and cathode of an electrochemical cell by coupling formaldehyde oxidation with hydrogen evolution, thus resulting in a reduced cell potential compared to water splitting. The scope of this manuscript is well aligned with Nature Communications, given the focus on an emerging system for efficient hydrogen production. Overall, the manuscript is thorough and well-written. The current densities achieved are significant and the high overall faradaic efficiency to hydrogen is notable. It should be mentioned, though, that coupling of organic molecule oxidation at the anode with hydrogen evolution at the cathode has been demonstrated previously as noted by the authors. Thus, the concept is not novel, but the demonstrated performance metrics represent a significant achievement. My major concern with the manuscript is around the interpretation and discussion/consideration of durability. Please see my specific comments below that I believe need to be addressed prior to further consideration:

1. If run in a continuous process under less dilute conditions, how does the pH of the electrolyte change over time in the anode and cathode chambers given the OH⁻ balance? How would this affect the durability of the process, especially considering moving beyond an H-cell into a scale-up electrochemical device?

Response: Following the reviewer's suggestion, controlled-current electrolysis at 20 mA was performed under dilute electrolyte conditions in a flow cell, in which catholyte (0.1 M KOH) and anolyte (0.1 M KOH with 0.6 M HCHO) were continuously fed into the cell at a flow rate of 5.0 mL min⁻¹. As shown in **Figure R16**, the catholyte showed negligible pH change during the electrolysis. The anolyte pH drop about 0.1 during the 1st hour and then remained quite stable for the rest of the electrolysis period. For future long-term operation using a scale-up electrochemical device, HCHO and KOH could be mixed on site and then fed into the anolyte chamber to minimize the impact of the Cannizzaro reaction.

Figure R16. pH change of anolyte and catholyte during electrolysis in a continuous flow process. Error bars represent the standard deviation of at least three independent measurements. The electrolysis was carried out at a constant current of 20 mA in a two-electrode flow cell using

the $\text{Cu}_3\text{Ag}_7/\text{CF}$ (anode) and $\text{Ni}_3\text{N}/\text{Ni}/\text{NF}$ (cathode) electrocatalyst couple. 0.1 M KOH was used as the catholyte while 0.1 M KOH with 0.6 M HCHO as the anolyte. Flow rates of both anolyte and catholyte were set as 5.0 ml min^{-1} .

2. The authors utilize Fig. 3E and 3F as evidence for the stability of the process and the electrocatalyst. However, in Fig. 3E, the slope of the voltage vs. time data at 500 mA/cm^2 is increasing quite considerably with each cycle (even with full replacement of the electrolyte). Further, in Fig. 3F, the maximum current density obtained decreases for each subsequent cycle. In both cases, this data is only collected over less than 10 hours, which is a short period of time relative to industrial processes. The authors need to discuss these results in detail and include interpretation about what it means for the long-term durability of this process, rather than just stating the cycles are “nearly identical”.

Response: For the stability tests of relatively short duration using a H-cell in the previous manuscript, the large amount of H_2 generated on the cathode and anode at a high current density could not escape in time, which might affect the mass transport of the electrolyte and substrate. Following the reviewer’s suggestion, longer electrolysis using a flow cell was performed. The long-term stability measurement of electrolysis was carried out at a controlled current density of 100 mA cm^{-2} in a flow cell, in which the catholyte (1.0 M KOH) and anolyte (1.0 M KOH and 10.0 g/L PFA) were fed into the cell at a flow rate of 50 mL min^{-1} . The electrolyte was refreshed every 24 h. Our dual H_2 production strategy (HER/FOR) using the $\text{Ni}_3\text{N}/\text{Ni}/\text{NF}(-) \parallel \text{Cu}_3\text{Ag}_7/\text{CF}(+)$ electrode couple could maintain a low voltage range of 0.4 – 0.6 V (without iR correction) over 260 h (**Figure R17**), indicating the excellent robustness of our electrocatalysts for this dual H_2 production system.

Figure R17. Chronopotentiometric curves for controlled-current electrolysis at 100 mA/cm^2 . The electrolysis was carried out at controlled-current electrolysis (100 mA cm^{-2}) in a two-electrode flow cell using the $\text{Cu}_3\text{Ag}_7/\text{CF}$ and $\text{Ni}_3\text{N}/\text{Ni}/\text{NF}$ couple. The catholyte (1.0 M KOH) and anolyte (1.0 M KOH and 10.0 g/L PFA) were fed into the cell at a flow rate of 50 mL min^{-1} . The electrolyte was refreshed every 24 h.

REVIEWERS' COMMENTS

Reviewer #1 (Remarks to the Author):

The authors responded well to the comments. This manuscript is ready for publication in this journal.

Reviewer #2 (Remarks to the Author):

The authors have carried out additional calculations and adequately extended the electronic structure analysis as claimed in the previous review. The computational results are consistent with the experimental observations. The manuscript can be published in its current state.

Reviewer #3 (Remarks to the Author):

In my opinion, in the revised version of the manuscript entitled "Dual H₂ Production from Electrocatalytic Water Reduction Coupled with Formaldehyde Oxidation via a Copper-Silver Electrocatalyst" the authors addressed the comments raised in an adequate manner and providing sound evidence when necessary. I thus support the publication of this manuscript.

REVIEWERS' COMMENTS

Reviewer #1:

The authors responded well to the comments. This manuscript is ready for publication in this journal.

Response: We are pleased that the reviewer was satisfied with our response. We thank the reviewer's valuable comments in improving the quality of our manuscript.

Reviewer #2:

The authors have carried out additional calculations and adequately extended the electronic structure analysis as claimed in the previous review. The computational results are consistent with the experimental observations. The manuscript can be published in its current state.

Response: We thank the reviewer for spending valuable time reviewing our manuscript. All the insightful comments from the reviewer are very helpful in making the manuscript of higher quality.

Reviewer #3:

In my opinion, in the revised version of the manuscript entitled "Dual H₂ Production from Electrocatalytic Water Reduction Coupled with Formaldehyde Oxidation via a Copper-Silver Electrocatalyst" the authors addressed the comments raised in an adequate manner and providing sound evidence when necessary. I thus support the publication of this manuscript.

Response: We appreciate the reviewer's suggestion of our manuscript for publication. We would like to thank again the reviewer's constructive suggestions for greatly improving the quality of this work.